# The Utilisation of Hydrogels for iPSC-Cardiomyocyte Research

**DOI:** 10.3390/ijms24129995

**Published:** 2023-06-10

**Authors:** Leena Patel, Joshua C. Worch, Andrew P. Dove, Katja Gehmlich

**Affiliations:** 1Institute of Cardiovascular Science, University of Birmingham, Birmingham B15 2TT, UK; lxp062@student.bham.ac.uk; 2School of Chemistry, University of Birmingham, Birmingham B15 2TT, UK; jworch@vt.edu (J.C.W.); a.dove@bham.ac.uk (A.P.D.)

**Keywords:** extracellular matrix, hydrogels, biomaterials, stiffness, cardiovascular

## Abstract

Cardiac fibroblasts’ (FBs) and cardiomyocytes’ (CMs) behaviour and morphology are influenced by their environment such as remodelling of the myocardium, thus highlighting the importance of biomaterial substrates in cell culture. Biomaterials have emerged as important tools for the development of physiological models, due to the range of adaptable properties of these materials, such as degradability and biocompatibility. Biomaterial hydrogels can act as alternative substrates for cellular studies, which have been particularly key to the progression of the cardiovascular field. This review will focus on the role of hydrogels in cardiac research, specifically the use of natural and synthetic biomaterials such as hyaluronic acid, polydimethylsiloxane and polyethylene glycol for culturing induced pluripotent stem cell-derived cardiomyocytes (iPSC-CMs). The ability to fine-tune mechanical properties such as stiffness and the versatility of biomaterials is assessed, alongside applications of hydrogels with iPSC-CMs. Natural hydrogels often display higher biocompatibility with iPSC-CMs but often degrade quicker, whereas synthetic hydrogels can be modified to facilitate cell attachment and decrease degradation rates. iPSC-CM structure and electrophysiology can be assessed on natural and synthetic hydrogels, often resolving issues such as immaturity of iPSC-CMs. Biomaterial hydrogels can thus provide a more physiological model of the cardiac extracellular matrix compared to traditional 2D models, with the cardiac field expansively utilising hydrogels to recapitulate disease conditions such as stiffness, encourage alignment of iPSC-CMs and facilitate further model development such as engineered heart tissues (EHTs).

## 1. Introduction

### 1.1. Cardiac Extracellular Matrix

The myocardium consists of multiple cell types [1], with cardiomyocytes and cardiac fibroblasts being the most abundant ones. In the healthy myocardium, the extracellular matrix (ECM) acts as a mechanical scaffold that provides structural organization and support to these cells and is key for maintaining cardiac homeostasis by mediating cellular responses [2]. Cardiac diseases such as myocardial infarction and cardiomyopathies trigger remodelling of the ECM, leading to increased collagen deposition and stiffness of the extracellular matrix as part of pathological changes of the myocardium [3,4]. The ECM is known to dictate cardiac cell behaviour by altering matrix rigidity, thus altering cell phenotypes and structure [5]. Thus, elucidating the matrix environment within the myocardium is relevant for understanding the progression of cardiovascular disease phenotypes. Recent cardiovascular research has shifted towards developing physiologically comparable synthetic ECM models that mimic healthy and diseased myocardium tissues, e.g., by utilizing hydrogels.

Induced pluripotent stem cell-derived cardiomyocytes (iPSC-CMs) are cardiac cells generated from stem cells which are widely used in the field for studying cardiac diseases and modelling cardiac phenotypes as alternatives to primary cells or tissue [6]. Many in vitro models use iPSC-CM and cardiac fibroblasts cultured on plastic dishes or glass, which often have a highly variable stiffness range from of 1 MPa–3 GPa. These conditions do not reflect physiological conditions in the heart, with the healthy ECM possessing a modulus of approximately 10 kPa whilst fibrotic ECM stiffness is approximately greater than 40 kPa [7,8]. The stiffness of cell culture surfaces often correlates to different disease phenotypes, such as transition of fibroblasts to myofibroblasts in cardiac fibrosis, which consequently activates distinct biomechanical pathways [9]. Therefore, cells cultured on stiffer substrates may not display a healthy phenotype, thus causing difficulty in distinguishing a ‘healthy’ phenotype for downstream comparison to diseased cells. Cell culture dishes, consequently, deviate from the biophysical environment, which highlights the requirement for softer substrates to examine cellular systems.

Several strategies have been employed to improve cell culture dishes and better mimic native myocardium environments. The dish surfaces are commonly coated with several substrates that are found in the ECM, such as collagen, laminin or fibronectin. Formulations that contain several of these ECM components, such as Matrigel or Geltrex, can further facilitate cell differentiation and attachment. However, a key drawback with these ECM formulation coatings is their batch-to-batch variability in composition and biomechanical properties (such as stiffness), which leads to a lack of experimental reproducibility [10]. Furthermore, the stability of ECM coatings can be problematic in long-term cell culture experiments. This is a notable issue when they are used for cardiac research, since cardiac cells such as iPSC-CMs only reach maturity after 30 days. The limitations associated with simple glasses or plastic dishes have driven the search for alternative substrates that possess greater structural integrity and are biologically, chemically and mechanically modifiable. 

### 1.2. Hydrogels as Synthetic Biomaterials Scaffolds

Biomaterials are natural or synthetic substances which can interact with biological systems and maintain signalling, thus playing a major role in tissue engineering advances [11,12]. Hydrogels, which are 3-dimensional crosslinked polymer networks that absorb and retain water whilst maintaining their structural integrity, have emerged as a promising class of synthetic biomaterials [13]. The structural integrity of hydrogels can be altered through chemical modification based on the crosslinking of polymer chains [14]. Specifically, the mechanical properties can be finely tuned by modulating several factors including polymer composition or charge, network architecture/crosslinking density, and responsiveness to external stimuli [15]. Furthermore, varying the composition and/or crosslinking density of hydrogels is an effective means to control their degradability or lifetime. The degree of crosslinking between polymers directly impacts the integrity of the hydrogel, with greater crosslinking resulting in a stronger gel but reduced flexibility of the network; a balance of network crosslinking is required to enable mechanical strength and elasticity of a hydrogel [16].

In addition to crosslinking, the molecular weight and chemical nature of polymers used for hydrogel synthesis can further impact the hydrophilicity and swelling behaviour of gels, as the incorporation of monomers that have high hydrophilicity portray greater swelling ratios [17]. Choice of polymer can therefore alter the formation and degradation properties of hydrogels. Due to the modular nature of hydrogels, they are extensively developed and applied as biomaterials for wide-ranging medicinal applications [18,19,20,21].

### 1.3. Functionalisation of Hydrogels

Hydrogel biomaterials often display poor cell adhesion properties; hence, functionalisation of hydrogels is key for transforming them from bioinert to bioactive. There are several methods of functionalising hydrogels to improve properties, with chemical modification being the most common. The installation of functional groups onto polymers or via post-functionalisation of the hydrogels after gelation can improve the adsorption capacity of hydrogels, in addition to refining their selectivity and stability [22]. In the search for more biocompatible reaction conditions, post-functionalisation strategies have moved away from inefficient metal-catalysed reactions and towards mild click reactions that employ organocatalysts [23]. 

Peptides, such as integrin binding peptides, are a common motif used to improve bioactive properties into hydrogels, where their chemical functionality promotes interaction between the hydrogel and cell components to facilitate cell adhesion [24]. The naturally occurring peptide arginyl-glycyl-aspartic acid (RGD) is most widely used for the functionalisation of gels to improve cell attachment (Figure 1) [25]. In general, the content of peptides in hydrogel scaffolds is positively correlated to the number of fibroblasts and cardiomyocytes attached to various hydrogels [26,27]. 

Electrostatic interactions also play a key role in functionalising hydrogels, as opposing charges between the surface of hydrogels and cells increase the strength of electrostatic attraction [22]. Charged residues are attracted to polar groups, and often these interactions occur at functional sites [28]. The electrostatic charges can facilitate interactions between enzymes, peptides and proteins and therefore improve cell adhesion to hydrogel surfaces. Mechanical properties of hydrogels and electrostatic interactions can also be influenced by various factors, such as the molecular weight of hydrogel materials, ratios of materials used, pH and temperature [29].

Various reactive groups on amino acids that are inherent (e.g., amines, thiols or carboxylates) or synthetically modified (e.g., azides) can be utilised to ligate peptides to hydrogels [30,31]. In particular, lysine and cysteine are commonly used amino acids within this context due to their accessible amino and/or thiol groups. For example, thiol-ene click chemistry has been commonly used for protein conjugation where the nucleophilic thiol or amino groups can react with electrophilic unsaturated bonds, such as alkynes or alkenes, under mild conditions to form robust thioether bonds [32].

Application of hydrogels in cardiovascular research

Hydrogels have been employed within in vitro cardiac research as they show great potential to represent the physiological conditions of the heart, by acting as tissue scaffolds that mimic the matrix environment. Consequently, most hydrogels for cardiac research are often derived from ECM proteins such as collagen, elastin or mixtures of ECM proteins such as Matrigel since they are naturally occurring and therefore may reflect an appropriate response to cells and remodelling [33]. In addition, the permeability of hydrogels allows for facile diffusion of nutrients, which is a desirable feature of the native ECM. 

Naturally occurring polymers, such as collagen, hyaluronic acid and alginate are a popular choice for developing hydrogels (Figure 2) since these polymers are inherently biocompatible and degradable [34]. However, natural hydrogels can suffer from a narrow property range (e.g., mechanical stiffness or degradability), very from batch to batch and are less amenable to covalent modification. In contrast, hydrogels that are based on synthetic polymers such as polyethylene glycol (PEG) are renowned for their high stability and modularity, with the polymers serving as ‘blank’ backbones for chemical modifications [35,36]. The overall structure of these hydrogels can be altered to access tailored properties, by simply adjusting the relative ratios of precursor polymers or even changing fundamental polymer characteristics such as molecular weight [37]. In addition, the mechanical stiffness of synthetic hydrogels can be modified through adjusting scaffold porosity and network interconnectivity including crosslinking density [38].

An essential feature when developing synthetic hydrogels is to ensure the maintenance of a desirable pH near 7.4 which imitates the ECM environment. Deviation from a physiological pH, often indicated through changes in colour of cell culture media, can result in acidic hydrogels, which are highly cytotoxic and thus incompatible for cell culture. This issue can arise when using synthetic hydrogels, where leaching of unreacted precursors or small molecules can change the pH or cause cytotoxicity. Therefore, it is often essential to thoroughly wash synthetic gels with physiological buffers or media to remove any remaining compounds/materials that are not incorporated into the network. The rest of this review will focus on examples of both natural and synthetic hydrogels that have been directly used in the field of cardiovascular research, whilst highlighting their respective strengths and weaknesses (Table 1).

## 2. Hydrogels from Natural Polymers

Natural hydrogels can further be classified into two primary groups, polysaccharides (such as hyaluronic acid and alginate) and proteins (such as collagen) [45].

### 2.1. Collagen/Gelatin Based Hydrogels

Collagen is the dominant extracellular protein in the heart, playing a key role in the structure of the cardiac myocardium. Gelatin, obtained by hydrolysis of collagen, is a water-soluble polymer (Table 1) that has been utilized widely for tissue engineering and hydrogel synthesis due to its ease of availability, low cost, biodegradability and cell attachment motifs and peptides [46,47]. In addition, gelatin can be easily modified and processed into various forms and thus is commonly used for hydrogel development and drug therapeutics [41]. Both collagen and gelatin hydrogels have high biocompatibility with iPSC-CMs, both as individual biomaterials and when blended with other proteins such as fibrin [48].

A key advantage of gelatin as a biomaterial is the lack of further ECM coatings required for cell adhesion; iPSC-CMs have been shown to attach directly to gelatin hydrogels for over 3 weeks in culture, which is ideal for cardiac cell ageing experiments [49]. Anisotropic alignment of sarcomeric structures in rat cardiomyocytes has also been portrayed with culture on gelatin gels [50]. Furthermore, cardiomyocytes have portrayed synchronized beating and calcium transients whilst pacing on gelatin hydrogels, indicating gelatin hydrogels are suitable for electrophysiological analysis [51].

Gelatin hydrogels recapitulate a range of soft stiffnesses with elasticities ranging from 1–236 kPa [50], which, however, limit the capabilities for producing stiffer hydrogels. In addition, gelatin hydrogels are more susceptible to degradation by proteases, increasing the degradation rate and decreasing the thermal stability of hydrogels [52,53]. Furthermore, the source from which collagen has been extracted to form gelatin can alter the stiffness and crosslinking of the hydrogel, thus impacting hydrogel properties and variability between gelatin hydrogels [54].

### 2.2. Alginate Gels

Alginate is a natural polymer (Table 1) that can be rapidly crosslinked in the presence of calcium, and it has been used extensively as a substrate to study cardiomyocytes [39]. A key feature of alginate hydrogels is their ionic crosslinking mechanism, which enables temperature independent gelation, facile cell encapsulation and good cell recovery [55]. Additionally, alginate has low toxicity and is readily available at a low cost. Furthermore, alginate hydrogels can range in stiffnesses, from 200–500 kPa (Table 1).

Alginate hydrogels with and without RGD peptide modification portrayed cardiac fibroblast attachment [56], highlighting the excellent cytocompatibility of alginate hydrogels with cardiac cells. Cardiomyocytes cultured on hydrogels formed from a combination of alginate and gelatin have shown successful beating of human iPSC-CM and sarcomeric protein expression [57]. Alginate scaffolds also hold advantageous electrophysiological properties, exemplified by their increased conductivity, enhanced cell alignment and higher expression of electrical coupling protein connexin-43 under electrical stimulation in cardiac cells [40]. 

Despite the maintenance of cardiac cell structure and beneficial electrophysiological properties of alginate hydrogels, there are several limitations of using alginate hydrogels in the cardiovascular field. The repelling anionic surface of native alginate hydrogels leads to issues with cell attachment, hence often requiring further functionalisation of alginate scaffolds or addition of peptides [58], which may further deviate from physiologically relevant conditions. Previous research and clinical trials using alginate scaffolds with cardiac fibroblasts have shown degradation over relatively short periods of time, varying from 15 to 21 days [59,60]. Cardiovascular research often requires long-term cultures of cardiac cells, such as iPSC-CMs which require a minimum of 30 days in culture. Thus, the rapid degradability of alginate hydrogels can be a significant challenge. 

### 2.3. Hyaluronic Acid-Based Hydrogels

Hyaluronic acid is a natural glycosaminoglycan (Table 1) that is abundant in the myocardial extracellular matrix and plays a role in cardiac development, structural organisation and wound healing processes [21]. The uncomplicated ability to modify hyaluronic acid hydrogels via peptide conjugation makes it an attractive biomaterial. Thiol-modified hyaluronic acid hydrogels portrayed increased dermal fibroblast attachment and proliferation up to 15 days in vitro cell culture when crosslinked with polyethylene glycol diacrylate and RGD peptides [61].

Furthermore, a range of stiffnesses are accessible starting from thiolated hyaluronic acid and varying the ratios and thus the crosslinking density, which is beneficial for testing different stiffnesses [62]. However, hyaluronic acid gels of stiffnesses greater than 1 kPa did not affect myofibrillar assembly of cardiomyocytes [63], indicating that the effect of stiffnesses may be saturated or overridden on hyaluronan gels and therefore may not serve as an accurate tool. Functionalisation of hyaluronic acid can also cause unintended issues, as gels synthesized with tyramine displayed cell leakage due to poor mechanical strength of the network coupled with low cell viability over 3 days [64]. As with alginate gels, studies using hyaluronic acid gels have highlighted issues with rapid degradation (thinning of gels) and reduced stiffness over time [65]. 

Cardiomyocyte maturity and alignment are essential factors in cardiovascular research, with several studies aiming to reflect physiological conditions by maintaining cardiac structures [66,67]. Hyaluronic acid, when added to neonatal rat cardiomyocytes and fibroblasts, disrupted the organisation of cells, with reduced alignment and increased tissue diameter [68]. The altered structure of cardiac cells when exposed to hyaluronic acid indicates significant issues with hyaluronic acid as a representative ECM model. Furthermore, injection of thiol hyaluronic acid hydrogels into rat myocardium led to an inflammatory response and recruitment of granulomas surrounding the hydrogel, ref. [69] suggesting the in vivo applicability of the hydrogels may be limited. The biocompatibility issues with hyaluronic acid hydrogels therefore highlight a key limitation of this substrate for investigation into cardiac cell function.

## 3. Synthetic Hydrogels

Compared to natural hydrogels, the use of synthetic hydrogels has steadily increased primarily due to their flexibility in controlling mechanical and chemical properties. Nevertheless, synthetic hydrogels often require additional derivatisation, usually via surface coating, to enhance cell attachment [70]. 

### 3.1. Polydimethylsiloxane

Polydimethylsiloxane (PDMS) is a silicone-based polymer (Table 1) that is commonly used in biological research across a wide-range of cell types due to its biocompatibility, optical clarity, low cost and flexibility in fabrication [71]. There are several commercially available formulations of PDMS; however, Sylgard 184 is the most prevalent precursor for hydrogel synthesis due to its simple curing and strong mechanical properties. Another significant strength of using PDMS hydrogels is the wide range of elasticity and mechanical stiffnesses that can be developed (Table 1). PDMS hydrogels can be created by blending Sylgard 184 elastomer curing agent and Sylgard 527 base agent and curing overnight at 65 °C. The different ratios of Sylgard 527 to Sylgard 184 alters the Young’s modulus of mechanical stiffness, and thus can achieve a range of elastic moduli from 12 kPa to 2.5 MPa [72]. The ability to combine PDMS hydrogels with other biomaterials such as hyaluronic acid and polyethylene glycol enhances mechanical properties and degradation, making it desirable for research [73,74].

The varying stiffnesses of PDMS have been utilised for studying fibrosis, as cardiac fibroblasts cultured on stiffened PDMS hydrogels portrayed activation and conversion to myofibroblasts [75], indicating that PDMS can mirror pathophysiological conditions of the ECM. In addition, other cardiac cell parameters such as cardiomyocyte electrophysiology can be assessed when using PDMS as a substrate. Large differences in calcium wave propagation velocity have been observed in ventricular myocytes cultured on stiff vs. soft PDMS, 2.7 MPa vs. 27 kPa [76]. Thus, this demonstrates that PDMS is suitable for mimicking both healthy and diseased cardiac environments. Further supporting evidence was found in increased contractility and calcium handling responses from iPSC-CMs cultured on PDMS substrates that possessed physiological stiffness [77]. 

Sarcomeric organisation and alignment of cardiac cells are features of successful differentiation and cell maturity, which are key factors observed when using alternative substrates for ECM environments. Micro-grooved PDMS hydrogels have been shown to improve iPSC-CM alignment (Figure 3), with greater organisation of sarcomeric structures [78] and higher expression levels of sarcomeric components compared to iPSC-CMs cultured on glass [79], further illustrating the benefits of employing PDMS hydrogels as an ECM substrate for cardiovascular research.

Despite the widespread use of PDMS in cardiac therapeutics, several limitations remain unaddressed. A key issue with untreated PDMS is the hydrophobicity of its surface, which negatively impacts protein adsorption and inhibits cell attachment [80]. Thus, ECM coatings (such as fibronectin or Matrigel) are often added to PDMS to enhance bioactive properties. However, delamination of the coating from the hydrogel surface is a significant issue, leading to increased cytotoxicity and apoptosis [81]. Furthermore, basal respiration and ATP production rates were found to be reduced in ventricular cardiomyocytes cultured on PDMS gels in comparison to gelatin hydrogels, independent of substrate rigidity [81]. This indicates PDMS may not portray metabolic changes of cardiomyocytes as well as other hydrogel substrates, which is an important factor to consider when reviewing candidates for ECM models. 

### 3.2. Polyacrylamide

Polyacrylamide (PAA) is a synthetic biomaterial (Table 1) that has widely been used for the study of cardiomyocytes, due to its non-toxicity, stability and hydrophilic properties [82]. The stiffness of PAA hydrogels (up to 200 kPa) can be easily modified by altering the ratio of bis-acrylamide crosslinker and acrylamide monomers [83], increasing the desirability of PAA hydrogels as ECM models.

A principal advantage of PAA hydrogels is that they are suitable for investigating electrophysiology of cells due to their comparatively low mechanical stiffnesses (Table 1). Patch clamping carried out on neonatal rat CMs cultured on 9 kPa PAA gels portrayed longer action potential durations than cardiomyocytes on softer (5 kPa) and stiffer (>15 kPa) PAA hydrogels [84]. The changes in CM electrophysiology based on substrate stiffness indicates PAA is also effective at mimicking the ECM. Furthermore, neonatal rat CMs cultured on 15 kPa PAA gels maintain sarcomere and F-actin structures [85]. Sarcomeric structure and myofibril alignment was also preserved in human iPSC-CMs cultured on 10 kPa polyacrylamide gels [67], thus highlighting the compatibility of PAA with several cardiac cell types. Cardiac fibroblasts cultured onto soft (2 and 4.5 kPa) PAA hydrogels and 2 kPa were shown to express α-smooth muscle actin after 15 days, indicating that the gels are good surrogates for native tissue as α-smooth muscle actin is a classic activated fibroblast marker [86]. PAA can also be used to investigate changes in gene expression (Figure 4).

Analogous to many other synthetic hydrogels, the non-ionic nature of polyacrylamide leads to inert surfaces and hence often demands functionalisation to promote effective adhesion of cells and proteins [87,88]. Furthermore, PAA hydrogels may not be suitable for simulating a fibrotic or ageing ECM, as PAA gels tend to have lower stiffnesses and and/or tensile strengths due to excessive swelling [89,90]. Furthermore, PAA hydrogels are often difficult to adhere to plastic surfaces, which restricts their more general use in cell culture studies [91]. In addition, the short life span of PAA hydrogels (usually up to 10 days) is not ideal for long-term iPSC-CM cultures [88,92]. Finally, PAA hydrogels often suffer from poor thermostability, with structural deterioration/degradation due to hydrolysis of amide groups, thus disrupting polymer chains [93].

### 3.3. Polyethylene Glycol (PEG)

Polyethylene glycol (PEG) hydrogels (Table 1) are widely used in the medical field as delivery strategies for bioactive molecules and tissue engineering, due to their biocompatibility, non-toxicity and propensity for chemical modification [94]. PEG holds several structural, physical and chemical properties that are similar to the ECM, such as its high-water solubility and level of hydration [95]. The well-developed chemistry behind PEG hydrogels offers potential to develop hydrogels with highly specific properties, which can be tailored to recapitulate the ECM. Despite PEG itself not having reactive functional groups along the polymer backbone, the end groups such as thiol and alkynes can be easily modified with reactive motifs to create telechelic polymers; low molecular weight polymers with functional groups on both chain ends are valuable hydrogel precursors [96]. This modularity enables the production of a vast range of hydrogels with various desirable gel properties. 

PEG hydrogels have been widely used in the cardiac field across several cell types. iPSC-endothelial cells encapsulated in PEG hydrogels have been shown to successfully incorporate into vascular networks and express genes involved in vasculature development [97]. Furthermore, PEG hydrogels can maintain cardiac cell structures, as 3D culture of adult murine cardiomyocytes encapsulated in PEG hydrogels have shown preserved sarcomeric integrity and t-tubular structure [44]. The lack of interference with cell structures signifies that PEG is a suitable material for assessing the effects of ECM environments on cardiac cell structure. In addition, contractility of aortic valve interstitial cells in PEG hydrogels has also been measured [98], indicating the potential of PEG hydrogels to assess a vast range of cardiac parameters such as stress fibre formation and molecular structure. However, a major limitation of PEG hydrogels is their distinctive lack of cell specific adhesion, which requires functionalisation (e.g., with RGD peptides) to increase their affinity for cell binding [30].

## 4. Conclusions

Cardiac research is highly dependent on the synthetic models that are used to explore cardiac diseases. Therefore, it is instrumental that these models are robust and physiologically relevant. Currently, many scientific in vitro findings are unable to be recapitulated in a whole organism. With the advancement in technologies such as the development of iPSC-CMs and induced pluripotent stem cell-derived fibroblasts, there is an unfulfilled need for a mechanically stable hydrogel that can withstand long term cell cultures and achieve a range of stiffnesses, to better imitate the ECM under various conditions and physiological states. Ideal hydrogel properties for iPSC-CMs would include limited degradability, biocompatibility, effective cell adhesion and low cost to enable wide-scale use. Chemical modification of hydrogels with attachment peptides such as RGD have enhanced iPSC-CM attachment to hydrogels, which are common issues experienced with iPSC-CMs due to the sensitive nature of the cells. Furthermore, iPSC-CMs cultured on hydrogels can enable maturation of cells, addressing a key issue of iPSC-CMs in the field. Another important consideration for hydrogels suitable for iPSC-CM culture include gels that do not interfere with ECM protein deposition, as iPSC-CMs in co-culture with FBs interact and deposit their own ECM proteins such as collagen [99,100]. Hydrogels that can coat surfaces such as coverslips and cell culture dishes would be beneficial, enabling cell culture compatibility and thus easing experimental design. 

Hydrogels offer a highly effective biomaterials platform, allowing for 2D cellular modelling in a more physiologically relevant environment than plastic cell culture dishes, and are better for emulating the adult myocardium and ECM. Hydrogels are highly versatile, inexpensive and biocompatible, characteristics which offer key improvements compared to traditional in vitro cell culture models. The ability to chemically modify hydrogels and change their network structure enables the creation of gels with highly tailored mechanical properties that are suitable for a range of applications within the cardiovascular field. Hydrogels can also be adapted to include conductive polymers, forming electroconductive gels capable of promoting regeneration of cardiac tissue and repair due to restoration of electrical coupling between cardiac cells, as found in the native myocardium. Furthermore, hydrogels can be employed as therapeutic strategies in 3D contexts, with cardiac cells encapsulated in hydrogels and injected into the body for regenerative purposes [101,102].

Other techniques can further enhance the fidelity of ECM models. For example, micropatterning of substrates can improve cell alignment, structural organisation and contractility, thus serving as a maturation strategy for iPSC-CMs [76,87,103]. Nano pillar electrode structures have also been used to improve maturity. A recent study reported a correlation between nanopillars and matured morphology of CMs through inducing differentiation via cytoskeleton rearrangement and F-actin assembly [104]. Nanopillar electrodes have also been utilised for iPSC-CM electrophysiological measurements, capturing intracellular action potentials across multiple cells [105]. Further models such as engineered heart tissue (EHTs) or engineered muscle tissue develop 3D culture systems by using a combination of cardiomyocytes and FBs to form 3D models of the myocardium, which are reflective of cardiac physiology [89,106].

## Figures and Tables

**Figure 1 ijms-24-09995-f001:**
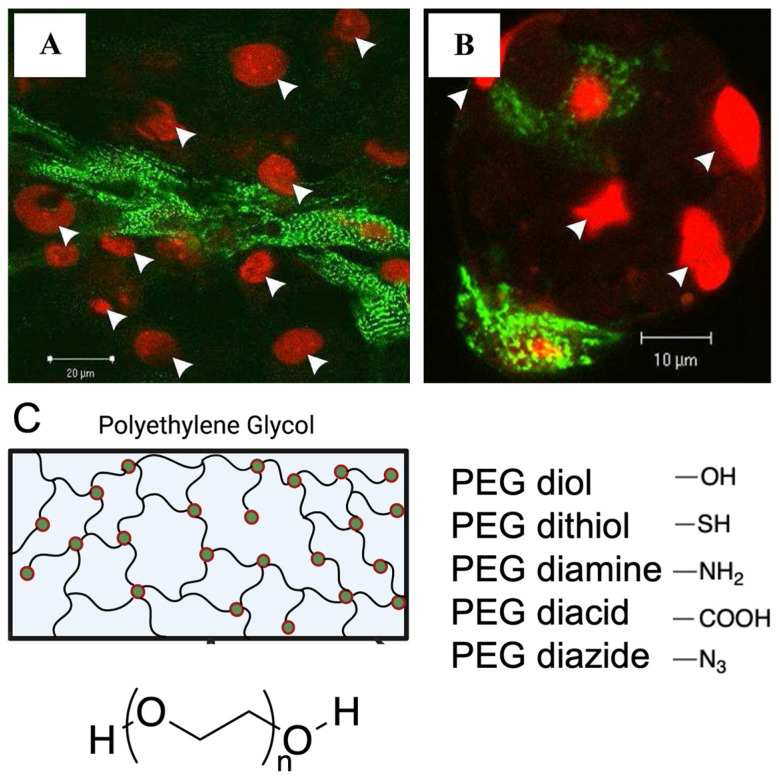
Functionalisation strategies, such as peptide modification and addition of functional groups to facilitate cell attachment onto synthetic hydrogels. (**A**,**B**). Attachment of rat ventricular cardiomyocytes to RGD modified alginate hydrogels compared to fewer cardiomyocytes attached to unmodified alginate hydrogels. Cardiomyocytes stained with α-actinin (green), nucleus stain with PI (red), white arrows indicate cell nuclei. Figure reproduced by Shachar et al., 2011 [27]. (**C**). Various end groups utilized for polyethylene glycol hydrogel functionalization.

**Figure 2 ijms-24-09995-f002:**
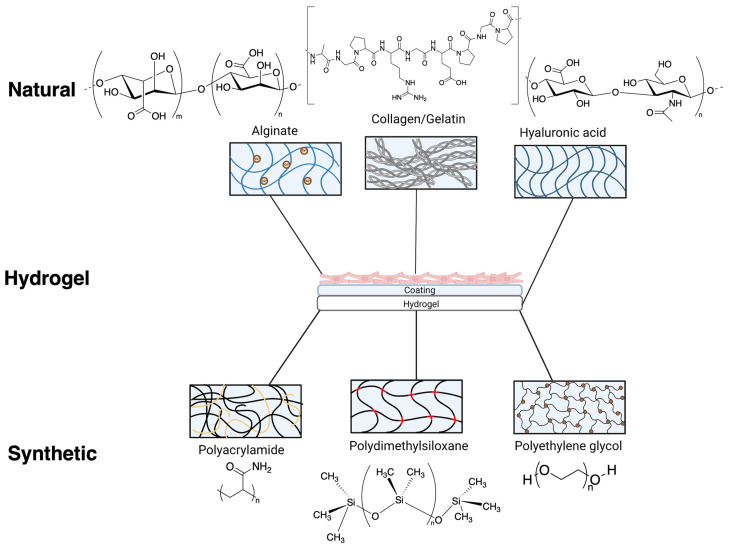
Various chemical composition/structure of natural hydrogels (formed from alginate, gelatin/collagen and hyaluronic acid) and synthetic hydrogels (formed from polyacrylamide, polydimethylsiloxane and polyethylene glycol) commonly used as substrates for cardiovascular research. Natural and synthetic hydrogels used for iPSC-cardiomyocytes often require additional coatings to facilitate attachment.

**Figure 3 ijms-24-09995-f003:**
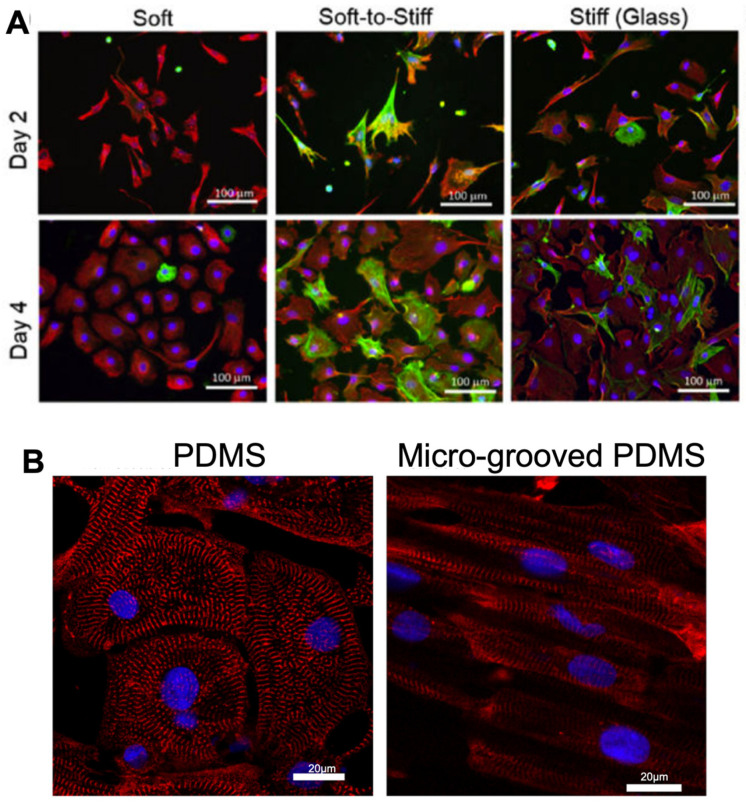
Morphology and structure of cardiac fibroblasts and cardiomyocytes altered on various stiffnesses of PDMS hydrogels, with micro-grooved PDMS improving alignment of iPSC-CMs. (**A**). Fibroblasts display activated morphology on glass compared to PDMS substrates of 2 kPa (soft PDMS), 37 kPa (Soft-to-stiff glass) and 1 GPa (Stiff glass) over 4 days in culture. Cells were stained with vimentin (a general marker for fibroblast, red), marker α-smooth muscle actin (a marker for ECM-producing activated fibroblasts, green) and DAPI visualizing nuclei (blue). Figure reproduced from Yeh et al., 2017 [75]. (**B**). iPSC-CMs display greater alignment when cultured on micro-grooved PDMS hydrogels vs unstructured PDMS, with sarcomeric marker α-actinin(red) and DAPI visualizing nuclei (blue). Figure reproduced from Rao et al., 2013 [78].

**Figure 4 ijms-24-09995-f004:**
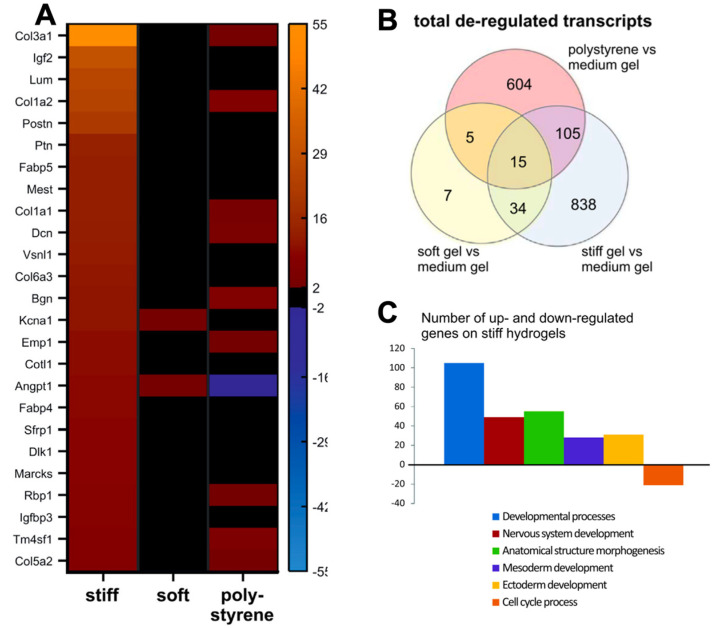
Transcriptomic profiles obtained from gene expression arrays of iPSC–CMs on polyacrylamide hydrogels of varying stiffnesses, portraying gene expression changes on hydrogels. (**A**). Gene expression profiles are altered when cells are cultured on PAA hydrogels of 12 kPa (soft) and 123 kPa (stiff), with increased expression of collagen proteins on stiffer substrates. (**B**). Venn diagram of gene ontology analysis of de-regulated pathways (**C**). De-regulated pathways from gene ontology analysis detected from CMs cultured on medium (30 kPa) and stiff (123 kPa) hydrogels. Figure reproduced from Heras-Bautista et al., 2019 [43].

**Table 1 ijms-24-09995-t001:** Overview of natural and synthetic and hydrogels properties. Range of stiffnesses reached for each material, with specific cases of biomaterial applications for iPSC-CMs in cardiac research, with overall advantages and disadvantages.

Hydrogel (Type)	Stiffnesses	Degradability	Applications	Advantages	Disadvantages
Alginate (Natural)	200–500 kPa	Degrades within 2 weeks	3D scaffold for ECM [39] Assessing electrical conductivity [40]	Electrical conductivity Low cost	Poor cell attachment
Hyaluronan (Natural)	1.5–8 kPa	5–7 days	Regenerative drug delivery [20]	Component of the ECM Fast gelation	Disruption of cell structure
Collagen/Gelatin (Natural)	1–236 kPa	Days-weeks, depending on temperature	Self-healing injectable therapeutic delivery [41]	Low cost Easily modified	Limited range of stiffness
Polydimethylsiloxane (PDMS) (Synthetic)	12 kPa–2 MPa	Can last up to several months	Assessing cell contractility [42]	Easy to produce Biocompatible	Hydrophobic
Polyacrylamide (PAA) (Synthetic)	1–200 kPa	Up to 10 days	Electrophysiology of iPSC-CMs [43]	Easily modifiable Optically transparent	Cell adhesion issues & lower stiffness range
Polyethylene glycol (PEG) (Synthetic)	10 kPa–1 MPa	Can be chemically modified to last months	Structural investigation of ventricular CMs [44] Injectable hydrogels improving cardiac function post MI [19]	Hydrophilic Non-toxic	Lack of cell specific adhesion

## Data Availability

Not applicable.

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
