# Peer review of "The Utilisation of Hydrogels for iPSC-Cardiomyocyte Research"

_ijms, 2023, doi:10.3390/ijms24129995_

Round 1

Author Response

Thank you for your comments and for taking the time out to review the paper, it is greatly appreciated. The following comments relating to specific lines are referring to the revised manuscript. 

1) Point one has been addressed by chaining the abstract. Lines 8-14 have been altered to be more concise and reduce background information. Lines 19-27 have been altered to add specific highlights of the review and future direction of biomaterial hydrogels in the field.

2) Other substrates used for iPSC-CMs such as Matrigel have been addressed on lines 63-72, with limitations of these substrates discussed and leading to explanations as to why hydrogels may be of greater potential. 

3) Lines 86-96 have been added to discuss how molecular weight and chemical cross linking of polymers can alter hydrogel properties and swelling of gels.  

4) Copyright permissions/Rightslinks for use of figures have been approved, references for figures and tables have been added to the list. 

5) Lines 426-436 have been added as specific guidelines for desirable properties of hydrogels designed for iPSC-CMs compatibility. 

Reviewer 2 Report

Patel et al. provide a review of the current options for hydrogels in use for cardiomyocyte research. Overall the written manuscript is of high quality and I have only a few suggestions.

1) I didn't see any mention of the pathological stiffness range is, or any quantitative comparison of stiffnesses.

2) The statement on line 83 is quite broad, should it have more than one reference?

3) I do not see a description of the colors for panels A and B in Figure 1.Also why are the scale bars different for the two images?

4) Do cells replace hydrogel with their own secreted ECM, and if so, are hydrogels available that degrade on a timescale that coincide with ECM secretion?

The quality of english is good, my only suggestion is a change to line 96:

Functionalisation strategies, such as peptide modification and additional of functional groups

Author Response

Thank you for reviewing the paper and leaving comments, the time you've taken is greatly appreciated! The line numbers indicated below are referring to the updated manuscript.

1) Lines 48-53 identify healthy myocardium stiffness and diseased/fibrotic stiffness, with a comparison to plastic stiffnesses currently used in the field. 

2) Multiple references have been added to Line 96

3) Colour descriptions have been added to Figure 1 in the caption. Parts A&B have been taken from a paper, thus the scale bars cannot be changed (reference number 24)

4) Cells can secrete their own ECM proteins, but this may not necessarily replace the hydrogel the cells are cultured on. The hydrogel reflects a surface for the cells to grow on and therefore shouldn't prevent a matrix from being formed. This has been mentioned on lines 432-434. Degradation of hydrogels can be altered, which has been discussed on lines 76-88.